# Easy Scheme Outlining the Various Morphological and Vascular Abnormalities of the Lymph Node Structure Associated with Recent COVID-19 Vaccination, Each with a Different Clinical/Diagnostic Management

**DOI:** 10.3390/jpm12091371

**Published:** 2022-08-25

**Authors:** Valeria Fiaschetti, Nicolò Ubaldi, Smeralda De Fazio, Elsa Cossu

**Affiliations:** 1Department of Biomedicine and Prevention, Tor Vergata University, 00133 Rome, Italy; 2European Hospital, 00149 Rome, Italy; 3Radiology Unit, Department of Medical Surgical Sciences and Translational Medicine, Sant’Andrea University Hospital, Sapienza University of Rome, 1035-1039, 00189 Rome, Italy; 4UOC of Diagnostic Imaging, Policlinico Tor Vergata (PTV), Tor Vergata University, 00133 Rome, Italy

**Keywords:** vaccination, lymphadenopathy, breast cancer, lymph node, biopsy, ultrasound

## Abstract

Throughout this recent ongoing SARS-CoV-2 pandemic, the European Society of Breast Imaging have surely contributed in improving the management of unilateral axillary adenopathy appearance homolaterally to the side of vaccine inoculation. After considering the patient’s COVID-19 history of vaccination, our group produced a day-to-day scheme that evaluates meticulously the probability of mammary malignancy, according to the lymph node characteristics including vascular abnormalities. It comprises of a UN (ultrasound node) score ranging from 2 to 5, that increases with the suspicion of malignancy. In this setting and in view of the additional incoming COVID-19 boost-dose vaccinations, we believe our model could be of great utility to radiologist when assessing patients whom do not have a straight forward diagnosis, in order to reduce breast cancer missed diagnosis, avoid delaying vaccinations, reduce rescheduling of breast imaging examinations and lastly avoid unnecessary lymph node biopsies.

## 1. Introduction

Since its outbreak back in December 2019 [1], SARS-CoV-2 has been a global burden; treatment of this disease has included mass vaccination campaigns. Since the first delivery of the COVID-19 vaccine in December 2020, at the time of writing (10 February 2022), 11.8 billion doses have been administered in total worldwide, generating a first dose coverage of circa 65% of the entire population. Of all the complications associated with the vaccine, we focus on axillary adenopathy, routinely identified during our prevention/screening program of breast cancer in clinical practice. It is now well-known that lymph node enlargement on the same side of the COVID-19 vaccine administration is a common inflammatory side effect; nevertheless, how a radiologist should interpret these image findings in a woman who has familiarity for mammary cancer, or, even worse, presents with a borderline Breast Imaging Reporting and Data System (BI-RADS) breast nodule, is an important question. In light of this, we present an alternative straight-forward workflow to help radiologists in their differential diagnosis reasoning.

Mammary cancer is considered the most common malignancy occurring in women, with over 2 million new cases in 2018 [2]. As axillary lymph nodes are located near the mammary gland, it is the first location to which malignancy can drain into. Therefore, axillary lymph node status is the main key prognostic factor to consider in metastatic breast cancer and should be used in therapy decision making. Interventional radiologists performing an ultrasound (US)-guided fine needle aspiration biopsy for suspicious axillary lymph nodes have shown a large proportion of positive metastatic lymph nodes identified pre-operatively [2,3]. In fact, sentinel lymph node biopsy is now widely accepted as axillary lymph node staging in patients with clinically negative nodes in order to reduce unnecessary lymph node dissections [3,4]. Up to now, the most authoritative institution working on breast cancer prevention: the American College of Radiology, in accordance with BI-RADS Atlas 5th edition, has stated that “in the absence of a known infectious or inflammatory cause, isolated unilateral axillary adenopathy [on screening mammography] should receive a BI-RADS category 0” [5,6]. Additionally, a unilateral lymph node enlargement at US examination with a clinical anamnesis of active infection in patients receiving COVID-19 vaccination should be treated as a BI-RADS 2 category finding. Nothing more than routinary mammography screening should be the correct path taken [5,6]. Lastly, “in the absence of a known infectious or inflammatory source, a suspicious (BI-RADS category 4) assessment would be appropriate” [5,6,7].

Moreover, between 14.5% and 53% of patients reported palpable axillary adenopathy after their first administered dose of COVID-19 vaccine, which persisted for >6 weeks in 29% [8]. Interestingly, COVID-19 vaccine administration in younger (<64 years) and immunocompetent patients was associated with a higher ipsilateral incidence of axillary lymphadenopathy compared to older and immunocompromised patients [7]. Of note, *Moderna* vaccine reported a higher incidence of ipsilateral axillary lymphadenopathy findings, accounting for 11.6% and 16% after the first and second dose, respectively [7]. Recently, the Society of Breast Imaging (SBI) and the European Society of Breast Imaging (EUSOBI) have elucidated the importance of a thorough anamnesis evaluation of the patient regarding the previous vaccination, in terms of: the inoculation site, any clinical variations and the timing related to the previous dose administration [9,10]. By doing so, the radiologist should be able to provide the best possible follow-up program for a unilateral axillary adenopathy. 

Importantly, one should be more precise when treating patients affected by breast cancer in order to decrease the number of false-positive axillary lymph node biopsy recommendations among other investigations and therapies to minimize patients’ harm and costs [9,10,11]. For these reasons, SBI and EUSOBI recommend that the most reliable way to carry out the correct breast cancer prevention program is either before a patient receives the COVID-19 vaccine or at least 1 month after [9,10]. The literature tells us that the average time of palpable axillary lymphadenopathy clinical resolution post COVID-19 vaccination ranges between 7 and 8 days, whilst evidence from PET-CT scan demonstrated that lymphadenopathy on imaging can persist even beyond 6 weeks [7]. Therefore, patients with ipsilateral axillary lymphadenopathy on US imaging and a history of receiving the COVID-19 vaccine within 1 month have two possibilities: Short-term follow-up with US imaging, starting from the fourth week after the second dose and up to the twelfth week.Persistent lymphadenopathy on follow-up imaging should prompt lymph node sampling in order to exclude malignancy [9,10,11].

We therefore produced an easy scheme outlining the various morphological and vascular abnormalities associated with the lymph node structure, each with a different clinical/diagnostic management. Our follow-up protocols in women presenting with axillary lymphadenopathy, both clinically and at imaging, is based on US findings subjective to radiologist expertise. We should differentiate suspicious malignant lymph nodes from benign ones and integrate the clinical history of COVID-19 vaccination in order to obtain the full picture pertinent to the patient’s health. 

## 2. Ultrasound Method in Assessing Lymph Node Morphological Features

Ultrasound can identify disease in a lymph node based on the presence or absence of defined sonographic criteria. US remains the most practical and efficient method, being free of ionizing radiation and the least costly modality to image the axillary station. With the dynamic nature and real-time imaging of ultrasound, there will always be variation secondary to operator expertise in detecting subtle differences that may subjectively class a node as abnormal [12,13]. A lymphadenopathy reaction can be found incidentally on imaging exams, such as routine screening or cancer surveillance (mammography, CT or MRI scans). The correct follow-up should be with US, as it is the preferred imaging modality for evaluating axillary lymph nodes [13,14]. Hence, a US examination will often be required. 

Morphologically, hilum features and cortical thickness of the lymph nodes are the most important criteria in order to distinguish between normal and abnormal lymph nodes. Further difficulty arises when attempting to classify the degree of abnormality; cortical thickening is often described as a suspicious feature, particularly when eccentric, though it is more difficult to define a threshold of concentric cortical thickness. Cortical thickness > 3 mm, round morphology and encroachment or displacement of the hyperechoic hilum are often suggestive of a pathologic process [12], and, together with the absence or replacement of fatty hilum, are a highly specific feature of nodal metastases [12,13,14,15]. The lymph nodes at US with round morphology and no hilum evidence, especially in patients under cancer surveillance, could be alarming [15]. Moreover, an additional majorly important factor when evaluating lymph node status is vascularization. On one hand a benign vascular pattern has been described as central vascularity, central perihilar vascularity or hilar vascularity, or the presence of a longitudinal vessel within the node (with or with- out branches) [16,17]. On the other hand, metastatic involvement of lymph nodes has frequently been associated with capsular or peripheral vascularity or with deformed radial and aberrant multifocal patterns [16,17].

A paper by Granata et al. [18], describing a population of 18 patients who received the Pfizer vaccine, found that 43.1% of lymph nodes showed eccentric cortical thickening with a wide echogenic hilum and oval shape, and 32.8% of lymph nodes showed asymmetric eccentric cortical thickening with a wide echogenic hilum and oval shape. A total of 17.2% of lymph nodes had a round or oval shape and showed concentric cortical thickening with a reduction in the width of the echogenic hilum, and 6.9% showed a huge reduction in and displacement of the echogenic hilum [18]. These results, from the current literature, demonstrate the heterogeneity of US features that can be found after COVID-19 vaccines and highlights the possibility of patterns mimicking malignancy [19,20]. 

To conclude, the lymph nodes ultrasound criteria that must be evaluated are:✓Morphology (oval vs. round shape/loss of bipolar ratio in diameter).✓Thickening of the cortex (>3 mm). ✓Color Doppler evaluation (centrally placed vascular hilum vs. small vessels entering cortex of node or no/aberrant vascular hilum).✓Adipose hilum (centrally vs. eccentric displacement fatty hilum or hilum absence).

We should pay attention to the morphology of lymph nodes; our group has produced a scheme assessing lymph node ultrasound structure, morphology and vascularization in order to meticulously assess the probability of malignancy (Figure 1). We created a scoring scale of the ultrasound node (UN) ranging from 2 to 5 (UN 2, UN 3, UN 4 and UN5): 2 being the least and 5 the most suspicious (Figure 1):
✓UN 2: Uniform cortex < 3 mm, centrally placed fatty and vascular hilum (color flow ultrasound), oval shape (preserved the long and short axis) (Figure 2). ✓UN 3: 3 mm cortex with uniform cortical thickness, centrally placed fatty and vascular hilum (color flow ultrasound), oval shape (preserved the bipolar ratio of diameters) (Figure 3). ✓UN 4: Localized bulge or uniform cortical thickness of cortex > 3 mm, eccentric displacement of fatty hilum, small vessels entering cortex of node (color flow ultrasound), rounded shape (loss of bipolar ratio of diameters) (Figure 4). ✓UN 5: Node with no fatty and vascular hilum or aberrant vascular patterns, globular shape (loss of bipolar ratio of diameters), irregular margins (Figure 5). 

Amonkar et al. [17] proposed a similar scheme, however without assessing lymph node vascularity. 

## 3. Differential Diagnosis and Management of Atypical Lymph Nodes

The management of patients with incidental unilateral axillary adenopathy identified during breast imaging after COVID-19 vaccination must be based on the clinical presentation: asymptomatic for screening, symptomatic breast and/or axilla for diagnosis, or recent breast cancer diagnosis in the pre-treatment phase [10,11]. It is important to know both when the patients were administered with the COVID-19 vaccine (first or second dose and dates received) and the side (left or right) and location (arm or thigh) of the inoculation. In the specific setting of screening mammography with no other findings beyond unilateral axillary adenopathy on the same side of the COVID-19 vaccination within 6 weeks from administration, we consider it a benign imaging finding and no further imaging is necessary [21,22,23]. Nevertheless, if there is clinical concern persisting for more than 6 weeks post-vaccination, axillary ultrasound repetition is recommended [21,22,23].

In this regard, SBI and EUSOBI, in order to avoid the diagnostic dilemma of vaccine-induced lymphadenopathy, advise to consider scheduling screening exams prior to the first dose or at least 4–6 weeks following the second dose of a COVID-19 vaccination [10,11]. The SBI and EUSOBI suggest collecting vaccination history on intake forms and educating patients that axillary swelling is a normal response to vaccinations. For subclinical unilateral axillary lymphadenopathy on mammary screening, the Society of Breast Imaging supports BI-RADS category 0, and it is advisable to bring patients back for further assessment of the ipsilateral breast and documentation of medical and vaccination history. BI-RADS category 3 is then accompanied with follow-up at 4 to 12 weeks after the second dose. BI-RADS category 4 (biopsy) should be considered when the lymphadenopathy persists on short-interval follow-up [10,11,15,18].

Adenopathy, identified incidentally on the side where vaccination occurred during diagnostic imaging workup for other breast signs or symptoms, is considered a BI-RADS category 2 if no suspicious findings are detected in the breast. In the setting of suspicious findings in the breast parenchyma (BI-RADS category 4 or 5), the management of the ipsilateral adenopathy is at the discretion of the attending procedural radiologist based on the grade of suspicion of the breast lesion, the appearance of the adenopathy, and pathology results [10,11,15,18]. For patients with breast cancer on the same side of the unilateral adenopathy after vaccination, core biopsy versus imaging or clinical follow-up is at the discretion of the breast surgeon and/or the medical or radiation oncologist in consultation with the radiologist [15,22,23]. The goal is to reduce unnecessary additional imaging and/or the biopsy of benign transient reactive axillary lymphadenopathy in the setting of recent ipsilateral deltoid muscle COVID-19 vaccination. 

We have previously seen how, theoretically, it is possible to manage patients with lymph node enlargement alone and with an associated mammary lesion. Nevertheless, in further detail, how patients with recent COVID-19 vaccine administration should be directed according to the morphology and characteristics of ipsilateral enlarged lymph nodes at US is another important question. Therefore, in the following section, we provide a more in-depth analysis of the overall management of these patients. This report will be one of the few in assessing in detail the characteristics of lymph nodes in terms of morphology and vascularization in patients with a previous COVID-19 vaccination. In light of the SBI and EUSOBI guidelines and the latest literature surveys, we propose a scheme to facilitate the management of daily unilateral axillary adenopathy findings post COVID-19 vaccination, identified incidentally throughout daily clinical practice (Figure 6).

Patients with recently documented (within the past 6 weeks) COVID-19 vaccination with axillary adenopathy in the ipsilateral arm reported by US but with no suspicious imaging finding in the breast and in the absence of a known infectious or inflammatory source:✓UN 3 score: Benign. No further imaging is indicated at this time.✓UN 4 or 5 score: US 6-week follow-up is recommended. If the lymph node has a lower score at the control, no further imaging is indicated at this time. If the score UN 4/5 remains, management is at the discretion of the attending procedural radiologist and/or surgeon based on laboratory or clinical evidence.

Patients with recently documented (within the past 6 weeks) COVID-19 vaccination with axillary adenopathy in the ipsilateral arm reported by US, with suspicious ipsilateral or contralateral breast cancer on mammography or ultrasound imaging:✓UN 3 score: The management is at the discretion of the attending procedural radiologist based on suspicion of lesion, pathology results, or a combination of these.✓UN 4/5 score: Collegial management must be taken (surgeon and/or medical or radiation oncologist in consultation with radiologist), evaluating the choice between fine needle aspiration cytology (FNAC)/biopsy vs. imaging or clinical follow-up.

In patients with a personal history of breast cancer and lymphadenopathy post-vaccination, nodal metastatic risk should be considered (cancer type, location, stage, etc.) [24,25]. 

✓UN 3 score:Unsuspicious breast finding requires short-interval follow-up imaging with ultrasonography (with at least a 6-week delay).Suspicious breast finding requires standard work-up, including FNAC/biopsy.✓UN 4/5 score: Node biopsy should be considered in the setting of high nodal metastatic risk and immediate histopathologic confirmation is necessary for timely patient management.

Radiologists should be aware that a recent COVID-19 vaccination can present an etiology of axillary lymph nodes with suspicious US features. Multiparametric ultrasound can be used as an additional aid, in particular the elastography [14,24].

Currently, we only have a limited amount of data from retrospective studies through collecting small and spontaneous samples; hence, we do not know the real incidence of ipsilateral axillary lymphadenopathy associated with COVID-19 vaccination.

## 4. Conclusions

Focusing on breast cancer prevention surveillance, we have seen how we can base our management approach on clinical history and presentation and vaccination administration and imaging findings at US. We have illustrated an easy scheme to follow in clinical practice to evaluate the morphology of lymph nodes in patients who received COVID-19 vaccination; compared to previous papers, we extended the analysis by integrating lymph node vascularity. This scheme comes with a general overview of morphological and volumetric differentiation between malignant and benign lymphadenopathy, which can also be of use outside the context of COVID-19 vaccination. Additionally, according to the lymph node morphological status, the clinical history and other specific breast findings, we proposed a management workflow that is reproducible in clinical practice. In this setting, we believe our model can help to reduce missed breast cancer diagnoses, avoid delaying vaccinations, reduce the rescheduling of breast imaging examinations and, lastly, avoid unnecessary lymph node biopsies. This approach could be of great utility, especially in view of additional incoming COVID-19 vaccinations.

## Figures and Tables

**Figure 1 jpm-12-01371-f001:**
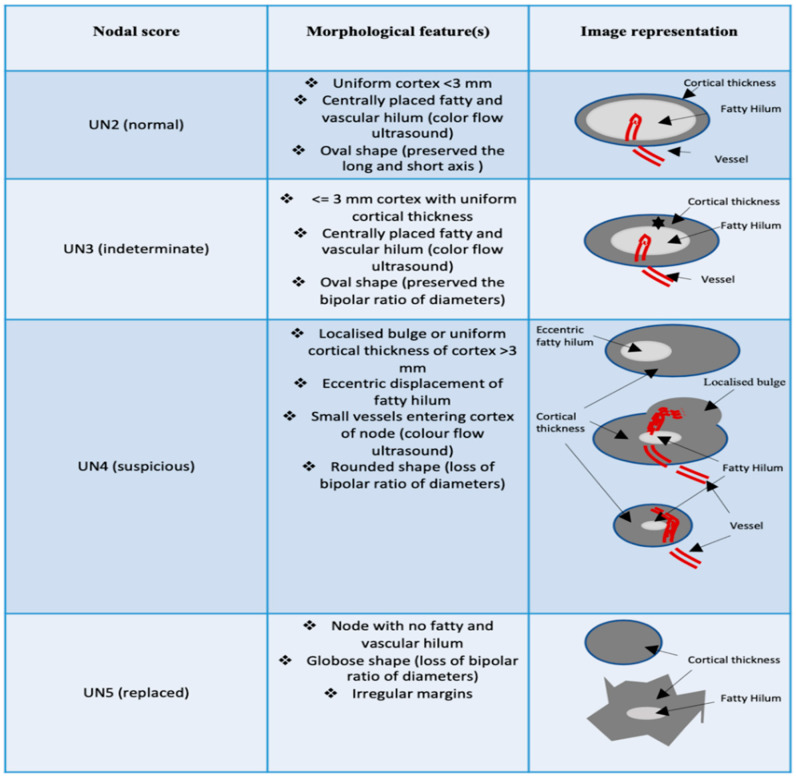
Allocated nodal score according to morphological and vascular features during ultrasound exam.

**Figure 2 jpm-12-01371-f002:**
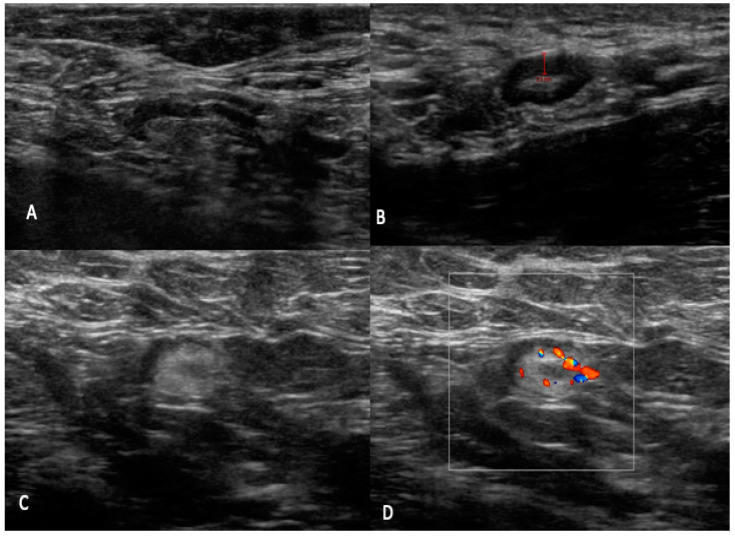
UN2: (**A**–**D**) B-mode sonogram image shows an oval node with uniform cortical thickening (<3 mm), centrally placed fatty and vascular hilum (color flow ultrasound).

**Figure 3 jpm-12-01371-f003:**
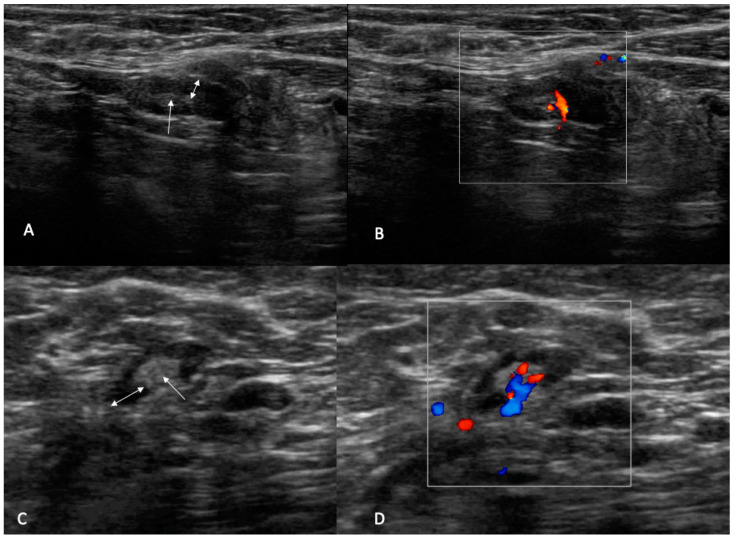
UN3: (**A**,**C**) B-mode sonogram image shows an oval node with uniform cortical thickening (double arrow) and centrally placed fatty (arrow). (**B**,**D**) Color flow images show central hilum single branch vascularization.

**Figure 4 jpm-12-01371-f004:**
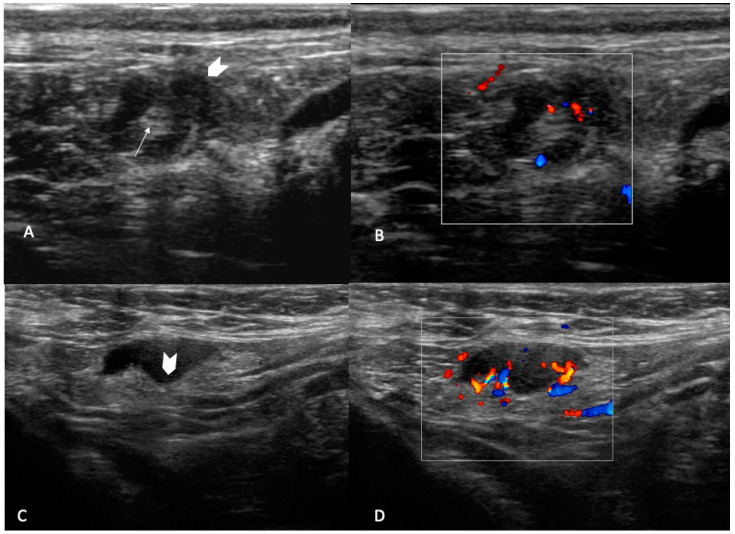
UN4: B-mode sonogram image shows a round, node with bulge external thickening (arrowhead) and centrally placed fatty (arrow) (**A**) and an oval node with bulge internal thickening (arrowhead) with displacement of fatty hilum (**C**). Color flow images show small vessels entering cortex of the node (**B**) and multiple branch vascularization (**D**).

**Figure 5 jpm-12-01371-f005:**
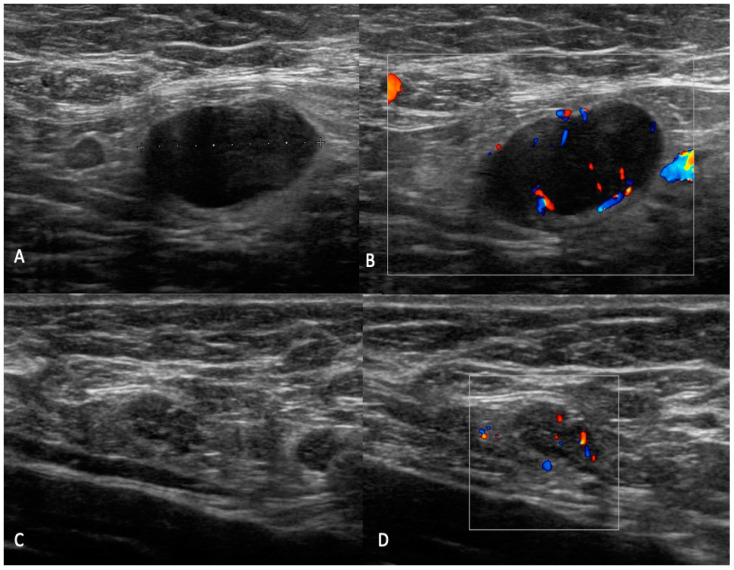
UN5: (**A**) B-mode sonogram image shows an enlarged hypoechoic lymph node with hilum absence. (**B**,**D**) Color flow images show aberrant vascular patterns with central and peripheral vascularization. (**C**) B-mode sonogram image shows a node with irregular margins, absence of fatty hilum and hyperechoic spots within.

**Figure 6 jpm-12-01371-f006:**
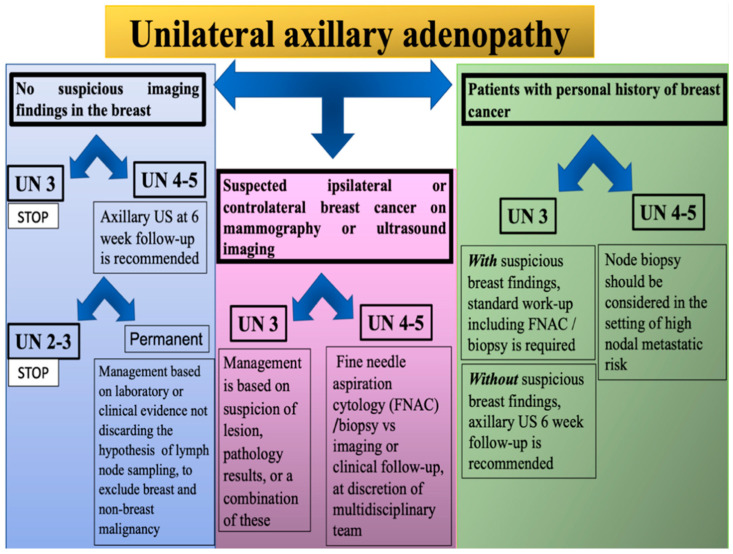
Management scheme of unilateral axillary adenopathy finding at ultrasound examination.

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
