# Peer review of "Easy Scheme Outlining the Various Morphological and Vascular Abnormalities of the Lymph Node Structure Associated with Recent COVID-19 Vaccination, Each with a Different Clinical/Diagnostic Management"

_jpm, 2022, doi:10.3390/jpm12091371_

Round 1

Reviewer 1 Report

Mild English editing is needed. I would advice to use a more concise language and improve quality of tables, specially table 2, but also table  . Figure 1 should be the same size as all others. Otherwise, the paper content is scientifically sound.

Author Response

Dear reviewer, 

We worked on the main file in review mode and tried to the best of our abilities to use more concise words.

We modified accordingly the size of all figures in order to keep them all the same. Also as proposed by the other reviewer, we decided to convert table 2 in figure 5.

Reviewer 2 Report

Authors aimed to review the various morphological and vascular abnormalities associated to the lymph node structure, including the SARS-CoV-2-induced axillary adenopathy. Therefore, authors presented a scheme outlining the the clinical/diagnostic management. However, some ponits should be revised:

- Although the pertinence of the topic, the "real" numbers of the cases of SARS-CoV-2-induced axillary adenopathy that progressed for breats cancer are not clear. My question is: are there any statistics ? 

Specific comments:

LIne 128 the sentence is not contextualized. 
Line 140 abreviation of Society of Breast Imaging. Please, check all abbrevaitions thoughtout the manuscript. 

I don´t understand the title of the first chapter. 

The text and figures formation are not in accordance with MDPI guidelines. 
I suggest to change table 2 for figure or diagram (please choose the more appropriate

The title should be changed to another one that reflects the main topic of the present mansucitp: "easy scheme outlining the various morphological and vascular abnormalities associated to the lymph node structure, each with a different clinical/diagnostic management"

The conclusions are not adequate. Please, reformulate. 

Author Response

Dear reviewer,

Response to 1st point:

We have not reported statistics regarding SARS-CoV-2-induced axillary adenopathy that progressed for breast cancer. This paper was intended to guide radiologists in the clinical practice rather than reporting statistics of determined clinical scenarios. 

Response to specific comments:

  • We have explained the reason behind sentence in line 128, this being it is to state that other authors e.g. Amonkar et al. have already proposed a similar diagram assessing the lymph node status but not they have not included lymph node vascularity.
  • We checked all abbreviations and they should be fine now.
  • We changed the title of the first chapter. The aim of this section was to highlight ultrasound method in evaluating lymph node's morphological features.
  • We modified the size of all figures in order to keep them all the same.
  • As also the other author suggested, we converted table 2 in figure 5.
  • We changed the title accordingly.
  • We modified the conclusion accordingly to the best of our abilities.